# Transthyretin Amyloid Cardiomyopathy: Impact of Transthyretin Amyloid Deposition in Myocardium on Cardiac Morphology and Function

**DOI:** 10.3390/jpm12050792

**Published:** 2022-05-13

**Authors:** Tomoya Nakano, Kenji Onoue, Chiyoko Terada, Satoshi Terasaki, Satomi Ishihara, Yukihiro Hashimoto, Yasuki Nakada, Hitoshi Nakagawa, Tomoya Ueda, Ayako Seno, Taku Nishida, Makoto Watanabe, Yoshinobu Hoshii, Kinta Hatakeyama, Yasuhiro Sakaguchi, Chiho Ohbayashi, Yoshihiko Saito

**Affiliations:** 1Department of Cardiovascular Medicine, Nara Medical University, Kashihara 634-8521, Nara, Japan; tnakano@naramed-u.ac.jp (T.N.); g_haxni_0922@naramed-u.ac.jp (S.T.); sassi0823@naramed-u.ac.jp (S.I.); y.hashimoto@naramed-u.ac.jp (Y.H.); ynakada@naramed-u.ac.jp (Y.N.); hitoshi.nakagawa@naramed-u.ac.jp (H.N.); tom15@naramed-u.ac.jp (T.U.); ayaseno@naramed-u.ac.jp (A.S.); taku99@naramed-u.ac.jp (T.N.); watamkt@naramed-u.ac.jp (M.W.); ysakaguc@wf6.so-net.ne.jp (Y.S.); yssaito@naramed-u.ac.jp (Y.S.); 2Department of Cardiovascular Medicine, Yamato Takada Municipal Hospital, Yamato-Takada 635-8501, Nara, Japan; 3Department of Diagnostic Pathology, Nara Medical University, Kashihara 634-8521, Nara, Japan; c.terada@naramed-u.ac.jp (C.T.); ohbayashi@naramed-u.ac.jp (C.O.); 4Department of Diagnostic Pathology, Yamaguchi University Hospital, Ube 755-0046, Yamaguchi, Japan; hoshii@po.cc.yamaguchi-u.ac.jp; 5Department of Pathology, National Cerebral and Cardiovascular Center, Suita 564-8565, Osaka, Japan; kpathol@ncvc.go.jp

**Keywords:** cardiac dysfunction, concentric left ventricular hypertrophy, endomyocardial biopsy, human heart tissue, immunostaining, reactive oxygen species, transthyretin amyloid cardiomyopathy

## Abstract

Background: Transthyretin (TTR) amyloid cardiomyopathy (ATTR-CM) is increasingly being recognized as a cause of left ventricular (LV) hypertrophy (LVH) and progressive heart failure in elderly patients. However, little is known about the cardiac morphology of ATTR-CM and the association between the degree of TTR amyloid deposition and cardiac dysfunction in these patients. Methods: We studied 28 consecutive patients with ATTR-CM and analyzed the relationship between echocardiographic parameters and pathological features using endomyocardial biopsy samples. Results: The cardiac geometries of patients with ATTR-CM were mainly classified as concentric LVH (96.4%). The relative wall thickness, a marker of LVH, tended to be positively correlated with the degree of non-cardiomyocyte area. The extent of TTR deposition was positively correlated with enlargement of the non-cardiomyocyte area, and these were positively correlated with LV diastolic dysfunction. Additionally, the extent of the area containing TTR was positively correlated with the percentage of cardiomyocyte nuclei stained for 8-hydroxy-2′deoxyguanosine, a marker of reactive oxygen species (ROS). ROS accumulation in cardiomyocytes was positively correlated with LV systolic dysfunction. Conclusion: Patients with ATTR-CM mainly displayed concentric LVH geometry. TTR amyloid deposition was associated with cardiac dysfunction via increased non-cardiomyocyte area and ROS accumulation in cardiomyocytes.

## 1. Introduction

Cardiac amyloidosis (CA) is an infiltrative secondary cardiomyopathy in which proteins with unstable tertiary structures misfold, aggregate, and form congophilic amyloid fibrils that deposit with a range of misfolded fibrillar proteins in the heart [1,2,3,4,5]. CA is characterized by left ventricular (LV) hypertrophy (LVH) due to the extracellular deposition of amyloid fibrils in the myocardium. Amyloid fibril deposition leads to LV dysfunction, heart failure with restrictive physiology, and eventual death [4,5,6]. More than 30 proteins can form amyloid fibrils, three of which frequently infiltrate into the myocardium and cause CA: immunoglobulin light-chain (AL), transthyretin amyloid (ATTR), and Amyloid A (AA) [2,4,7]. Transthyretin amyloid cardiomyopathy (ATTR-CM), the major cause of CA, is categorized into inherited mutations (variant ATTR-CM (ATTRv-CM)) and the aging process without mutations (wild type ATTR-CM (ATTRwt-CM)). Recently, ATTRwt-CM is increasingly recognized as a cause of progressive heart failure, restrictive cardiomyopathy, or LVH in elderly patients [2,5,6,8]. Furthermore, 13% of elderly patients (mean age, 82 years) who have heart failure with preserved LV ejection fraction, 16% of elderly patients (mean age, 83.7 years) with severe aortic stenosis undergoing transcatheter aortic valve replacement, and 8.2% of patients (mean age, 60 years) initially diagnosed with hypertrophic cardiomyopathy, were all reported to have a diagnosis of ATTR-CM [9,10,11,12,13]. Although ATTR-CM has been reported to manifest as concentric LVH [4], it is not clear whether this geometry is applicable to a greater number of cases with biopsy proven ATTR-CM.

Transthyretin (TTR) is known as prealbumin, and its function is to transport thyroxine and retinol-binding protein. TTR is synthesized mainly by the liver and circulates as a homotetramer [14]. In ATTR-CM, fibrillogenesis requires the dissociation of the TTR homotetrameric structure into misfolded monomers that self-assemble into soluble oligomeric species, which then misassemble into amyloid fibrils and deposit in the myocardium [4,5,6,14,15]. Previous studies have shown that TTR amyloid deposition is closely related to reactive oxygen species (ROS) accumulation and cell apoptosis [16,17]. Several clinical studies have reported that both stabilizers of TTR homotetramers and RNA interference therapies inhibiting TTR synthesis prevent the progression of ATTR-CM [15,18,19,20,21]. Therefore, the differentiation and early diagnosis of ATTR-CM in hypertrophic and failing hearts have become more important than ever. Furthermore, the association between the degree of TTR amyloid deposition in the myocardium and cardiac function remains unclear.

In this study, to identify the LVH geometry and investigate the mechanism of LV dysfunction in patients with ATTR-CM, we evaluated the pathological features, cardiac morphology, and function using human endomyocardial biopsy samples.

## 2. Materials and Methods

### 2.1. Patient Population

We included 40 consecutive CA patients diagnosed with endomyocardial biopsy at Nara Medical University Hospital between January 2007 and March 2021. They underwent coronary angiography and left ventriculography to exclude coronary artery and valvular diseases. Endomyocardial biopsy samples were obtained from a standardized location at the apical posterior wall of the left ventricle or the interventricular septum of the right ventricle. Among the 40 consecutive CA patients, 28 (70.0%) were diagnosed with ATTR-CM based on immunostaining, as described below, and were enrolled in this study. The remaining 12 patients had been diagnosed as CA other than ATTR-CM and were excluded from this study; one had Ig-κ AL cardiomyopathy (AL-CM), eight had Ig-λ AL-CM, and three had AA cardiomyopathy. In addition, an analysis of the *TTR* gene was performed for 17 consenting patients with ATTR-CM. All exon regions of the gene were sequenced using Sanger sequencing.

The study protocols were approved by the Nara Medical University Ethics Committee (protocol codes 1377 and G107, dates of approval: 17 October 2016 and 10 February 2014) and followed the 1975 Declaration of Helsinki guidelines. Written informed consent for the approval of patient information analysis and publication was obtained from either the patient or his/her family members.

### 2.2. Clinical Assessment

Clinical data, including electrocardiography (ECG) and transthoracic echocardiography (TTE) parameters, comorbidities, oral medications, and laboratory data, were obtained on admission. The estimated glomerular filtration rate (eGFR) was calculated according to the following equation for Japanese subjects [22]: 194 × serum creatinine^−1.094^ × age^−0.287^ (×0.739 for women). In evaluating the ECG, low QRS voltage was defined as an amplitude <0.5 mV in all limb leads. Poor R-wave progression in the precordial leads was defined as the loss or absence of R waves in V1-3. Atrial fibrillation included a history of paroxysmal atrial fibrillation and sustained atrial fibrillation. As for the parameters of TTE, LV ejection fraction (LVEF) was calculated via the modified Simpson method. Interventricular septum (IVS) thickness, posterior wall (PW) thickness, LV end-diastolic diameter (LVEDD), and LV end-systolic diameter were measured using M-mode. The early transmitral filling velocity (E) was measured using pulsed-wave Doppler. Using tissue Doppler imaging in the apical four-chamber view, the peak early diastolic velocity of the septal mitral annulus (e’) was measured, and the E/e’ (septal) ratio was calculated. Patterns of LV geometry were defined according to the American Society of Echocardiography recommendations [23]. Relative wall thickness (RWT) was calculated as 2 × (PW thickness)/LVEDD. Left ventricular mass (LVM) was calculated as 1.04 × [(LVEDD + IVS thickness + PW thickness)^3^ − LVEDD^3^] × 0.8 + 0.6 (g). LVM was normalized to body surface area (LVM index). Concentric LVH geometry was defined as an increased LVM index (>115 g/m^2^ in men and >95 g/m^2^ in women) with RWT > 0.42. Concentric LV remodeling geometry was defined as having a normal LVM index (≤115 g/m^2^ in men and ≤95 g/m^2^ in women) with RWT > 0.42. Eccentric LVH geometry was defined as an increased LVM index with RWT < 0.42.

### 2.3. Histological Assessment

Endomyocardial biopsy samples were fixed with 10% buffered formalin, embedded in paraffin, and sectioned at 4-µm thickness. Specimens were processed for hematoxylin-eosin (HE) and Masson’s trichrome (MT) stain. The cardiomyocyte diameter was measured at the nuclear level in at least 20 randomly selected cardiomyocytes from HE staining in a high-power field using a BZ-X710 microscope and BZ-X analyzer software system (BZ-X710 system, Keyence, Osaka, Japan). CA was diagnosed based on the pathological information. When CA was suspected, based on HE and MT staining, we added Congo red stain. If Congo red was observed within the interstitial area, immunostaining to distinguish the amyloid subtype was performed using anti-TTR, immunoglobulin kappa (Igκ), immunoglobulin lambda (Igλ), and AA antibodies (conducted at Yamaguchi University, Ube, Japan). A polyclonal antibody against a synthetic peptide corresponding to positions 115–124 of TTR (1:750 dilution) [24], polyclonal antibodies against a synthetic peptide corresponding to positions 116–133 of Igκ light chain (1:600 dilution), and positions 118–134 of Igλ light chain (1:600 dilution) [25], and a monoclonal antibody against AA amyloid (GA60561-2, 1:750 dilution; DAKO, Glostrup, Denmark) were used. We also studied the immunostaining of 8-hydroxy-2′-deoxyguanosine (8-OHdG) (ab48508, 1:200 dilution; Abcam, Cambridge, United Kingdom) to examine ROS production. Detailed methods are described in Appendix A.

### 2.4. Immunohistochemical Analysis

We assessed the association between echocardiographic parameters and histological features obtained by TTR immunostaining in patients with ATTR-CM. To cover the entire myocardium area in the high-power field, multiple images were captured. These high-resolution images were jointed using the BZ-X analyzer software. The total myocardium area and the areas showing TTR were measured using TTR immunostaining with the BZ-X710 system (Appendix A). Endocardial and vascular structures were excluded from the total myocardial area. The area occupied by cardiomyocytes was measured using autofluorescence of the same TTR immunostaining sample (Appendix A). The non-cardiomyocyte area, which indicated the extracellular region, was calculated as the difference between the total myocardial area and cardiomyocyte area. The non-cardiomyocyte area and the area containing TTR were calculated as percentages of the total myocardial area. Furthermore, we examined the involvement of ROS production in the mechanism of cardiac dysfunction in ATTR-CM. Cardiomyocyte nuclei stained with 8-OHdG were assessed as a percentage of the total number of cardiomyocyte nuclei.

### 2.5. Statistical Analysis

Continuous variables were expressed as means and standard deviations or medians and interquartile ranges (IQR), as appropriate. Categorical variables were expressed as percentages. Spearman’s correlation test was used to determine the relationship between the two variables. Statistical significance was set at *p* < 0.05. JMP software (SAS Institute, Cary, NC, USA) was used to analyze the data.

## 3. Results

### 3.1. Clinical Characteristics and LV Geometries of the Patients with ATTR-CM

The baseline clinical characteristics of the 28 patients with ATTR-CM are presented in Table 1. Mean age was 76.2 ± 6.4 years, and 26 patients (92.9%) were men. The mean LVEF was 52.4 ± 10.6%, and mean E/e’ (septal) ratio was 21.7 ± 7.1, mean IVS thickness was 15.6 ± 2.7 mm, mean PW thickness was 15.1 ± 2.3 mm, and mean LVEDD index was 44.5 ± 5.6 mm. The median brain natriuretic peptide (BNP) level was 298.1 pg/mL and the median high-sensitivity cardiac troponin T was 72.5 ng/L. None of the patients had taken either stabilizers of TTR homotetramers or RNA interference therapies inhibiting TTR synthesis. No mutations in the TTR gene were found in 17 consenting patients.

We analyzed the distribution of the LVM index and RWT among the 28 ATTR-CM patients by sex (Figure 1). Among the 28 patients with ATTR-CM, 27 (96.4%) had concentric LVH geometry, one (3.6%) had concentric LV remodeling geometry, and no patients had eccentric LVH or normal geometries.

### 3.2. Relationship between Pathological Features and Cardiac Morphology and Function

Representative general staining of endomyocardial biopsy specimens from patients with ATTR-CM is shown in Appendix A. We assessed the pathological features and their association with clinical parameters among 28 ATTR-CM patients; for 27 patients, the endomyocardial biopsy was from the LV, and for one patient from the RV.

The measurements of the degree of the area that was stained for TTR, the non-cardiomyocyte area, and the cardiomyocyte area were represented in Table 2.

The proportion of the area that was stained for TTR positively correlated with that of the non-cardiomyocyte area (*p* < 0.001), as shown in Figure 2A. However, the proportion that was TTR positive did not correlate with the cardiomyocyte diameter (Appendix A). The size of the TTR area correlated with the E/e’ (septal) ratio (*p* = 0.03; Figure 2C), but not with LVEF or RWT, which is a marker of LVH (Figure 2B,D). Although the extent of the non-cardiomyocyte area was not associated with LVEF (Figure 2E), it was positively correlated with the E/e’ (septal) ratio (*p* = 0.01; Figure 2F) and tended to have a positive correlation with RWT (*p* = 0.06; Figure 2G). Cardiomyocyte diameter was not correlated with the size of the non-cardiomyocyte area, LVEF, E/e’ (septal) ratio, or the RWT (Appendix A).

### 3.3. Association between ROS Accumulation and Pathological and Clinical Features

We also performed immunostaining of 8-OHdG to examine ROS accumulation in patients with ATTR-CM (Figure 3A). The association between the degree of ROS accumulation and TTE parameters was assessed. The size of the area exhibiting TTR correlated positively with the fraction of 8-OHdG stained cardiomyocyte nuclei (*p* = 0.01) (Figure 3B). The fraction of 8-OHdG stained cardiomyocyte nuclei were positively correlated with the percentage of the non-cardiomyocyte area and cardiomyocyte diameter (*p* = 0.03, and *p* = 0.04, respectively; Figure 3C and Appendix A). The fraction of 8-OHdG stained cardiomyocyte nuclei were negatively correlated with LVEF (*p* = 0.007; Figure 3D) but was not correlated with the E/e’ (septal) ratio or RWT (Figure 3E,F).

## 4. Discussion

The present study of patients with ATTR-CM demonstrated the following significant novel findings: (1) 96.4% of patients had concentric LVH geometry; (2) enlargement of the non-cardiomyocyte area, with an increased amount of TTR amyloid deposition, was associated with LV diastolic dysfunction and LVH; (3) TTR amyloid deposition and enlargement of the non-cardiomyocyte area affected the amount of ROS accumulation in the cardiomyocytes; and (4) ROS accumulation correlated with LV systolic dysfunction.

### 4.1. Cardiac Morphology

The clinical phenotypes of ATTR-CM include symmetrical wall thickening and progressive heart failure [4,5,6]. LV wall thickness is generally more prominent at diagnosis in patients with ATTR-CM than in patients with AL-CM, because patients with AL-CM tend to become symptomatic even in the early phase of their disease course [3]. Furthermore, LV wall thickening of the ATTR-CM heart starts from the basal region and progresses to the apex because LV myocardial amyloid infiltrates from the basal region [12]. In this study, most patients (96.4%) were categorized as having concentric LVH geometry. It has been reported to be the major morphological abnormality in patients with ATTR-CM with LV wall thickness and an increase in LV mass; however, LV dimension does not increase as the disease stage proceeds [3,26,27]. Conversely, patients with dilated cardiomyopathy (DCM) have increased LV dimensions to maintain cardiac output and demonstrate eccentric LVH geometry. For the patients with ATTR-CM, the LV geometry could not dilate, probably because of the stiffness of the LV wall. Consequently, patients with advanced ATTR-CM often demonstrate a restrictive pattern with severe impairment of LV filling and LV diastolic function.

Furthermore, in this study, one patient with ATTRwt-CM with concentric LV remodeling geometry demonstrated a lower BNP level (32.6 pg/mL) without diuretic usage and better LV systolic and diastolic functions, compared to 27 patients with ATTR-CM with concentric LVH geometry, indicating that this patient might be in the early stage of disease development [28]. Lai et al. reported that the geometry of patients with ATTRv-CM becomes more severe with progression from normal to concentric remodeling and concentric LVH as the disease stage proceeds [27]. The present study suggests that the transition of geometry in patients with ATTR-CM was consistent with that in patients with ATTRv-CM. Therefore, to prevent overt heart failure, we must diagnose ATTR-CM before patients manifest concentric LVH geometry, the most advanced ATTR-CM phenotype.

### 4.2. Relationship between the Pathological Features and Cardiac Dysfunction

The combination of TTR amyloid deposition and interstitial fibrosis contributes to the formation of non-cardiomyocyte areas [29], and TTR amyloid deposition accounts for a large portion of the non-cardiomyocyte area. A previous study demonstrated that the dysfunctional clearance of TTR non-fibrillar aggregates by cardiac fibroblasts, resulting from the inability of lysosomes to deal with the intracellular load of these aggregates, is involved in increased TTR amyloid deposition [30]. In addition, cardiomyocyte damage and microvascular dysfunction through TTR amyloid deposition are involved in the formation of interstitial fibrosis [31]. Our study suggested that TTR amyloid deposition was significantly associated with the extent of the non-cardiomyocyte area.

Furthermore, there is little understanding of the association between the pathological features and LV diastolic dysfunction in ATTR-CM. This study showed that enlargement of the non-cardiomyocyte area, but not cardiomyocyte hypertrophy, was associated with LVH. Our findings suggest that enlargement of the non-cardiomyocyte area together with an increased amount of TTR amyloid deposition might result in increased stiffness of the LV wall and consequent LV diastolic dysfunction in ATTR-CM hearts.

### 4.3. The Involvement of ROS Accumulation and Cardiac Dysfunction

In this study, using endomyocardial biopsy samples, we found that LV systolic dysfunction was associated with the extent of ROS accumulation in cardiomyocytes, but was not associated with TTR amyloid deposition or the non-cardiomyocyte area. In this study, the degree of TTR amyloid deposition and the non-cardiomyocyte area was positively associated with ROS accumulation in cardiomyocytes. Previous reports have demonstrated that mature TTR amyloid fibrils do not cause cellular damage, but TTR non-fibrillar aggregates are toxic to cardiomyocytes [32,33,34,35]. TTR non-fibrillar aggregates induce a cytotoxic response that involves mitochondrial dysfunction, endoplasmic reticulum stress, and dysregulation of the Ca^2+^ balance. Consequently, these cytotoxic responses may result in ROS accumulation and inflammation progression [17,33,34,35,36,37,38,39,40,41].

Additionally, it is well known that ROS accumulate by activation of neurohormonal systems, including the sympathetic nervous system and renin-angiotensin-aldosterone system [41,42,43,44,45]. In this study, the plasma renin activity and aldosterone concentrations were higher than the normal range. In patients with ATTR-CM, it has been reported that neurohormonal systems might be activated to increase cardiac output, since the ability to increase preload is limited because of severe impairment of LV filling [12,35,46,47].

### 4.4. The Mechanism of Cardiac Dysfunction in the ATTR-CM Heart

TTR amyloid deposition and subsequent enlargement of the non-cardiomyocyte area might contribute to LVH and LV diastolic dysfunction. ROS accumulation through the increase of TTR amyloid deposition and non-cardiomyocyte areas might be more sensitively related to LV systolic dysfunction. Based on our findings, the predicted mechanism of cardiac dysfunction in ATTR-CM hearts is shown in Figure 4.

Stabilizing TTR homotetramers or reducing TTR production might preferably be applied early in the stage before TTR amyloid fibrils accumulate in the myocardium. This approach is necessary because the severity of TTR deposition was related to the extent of the non-cardiomyocyte area and was thought to affect cardiomyocyte damage. The fact that tafamidis, a stabilizer of the TTR homotetramer, was shown to have a better response in treating patients classified as New York Heart Association (NYHA) functional class I and II compared with those classified as NYHA class III supports this concept [19]. Early diagnosis and management of ATTR-CM could help prevent cardiac dysfunction by suppressing amyloid deposition and ROS accumulation.

### 4.5. Limitations

This study has some limitations. First, the sample size was small and all samples were obtained from a single hospital. Second, the proportion of patients with ATTR-CM having concentric LVH geometry might be higher since endomyocardial biopsies might be mainly performed on patients with clinical symptoms in the advanced stage of the disease. Third, not all patients with ATTR-CM underwent genetic analysis. Fourthly, echocardiographic parameters of subclinical systolic dysfunction, such as global longitudinal strain, were measured in only four patients in this study. Given that most patients with ATTR-CM had preserved LVEF, further investigation using speckle tracking analysis is needed to elucidate the relation between the TTR deposition and cardiac dysfunction. Fifthly, the endomyocardial biopsy technique may have sampling errors in establishing a pathological diagnosis, although endomyocardial biopsy is reported to have a higher positive probability of detecting amyloid deposition than other methods, such as abdominal fat tissue biopsy in patients with ATTR-CM. A larger sample size should be evaluated to further clarify the mechanisms underlying ATTR-CM development.

## 5. Conclusions

ATTR-CM mainly exhibited concentric LVH geometries. TTR amyloid deposition is associated with cardiac dysfunction via increasing the areas without cardiomyocytes and promoting ROS accumulation in cardiomyocytes.

## Figures and Tables

**Figure 1 jpm-12-00792-f001:**
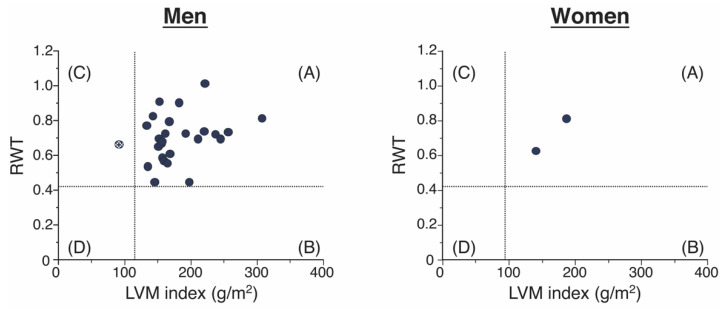
The figure shows the distribution of patient results according to the four geometries of the American Society of Echocardiography [23]: (**A**) concentric LVH, (**B**) eccentric LVH, (**C**) concentric LV remodeling, and (**D**) normal. LV, left ventricular; LVH, left ventricular hypertrophy; LVM, left ventricular mass; RWT, relative wall thickness.

**Figure 2 jpm-12-00792-f002:**
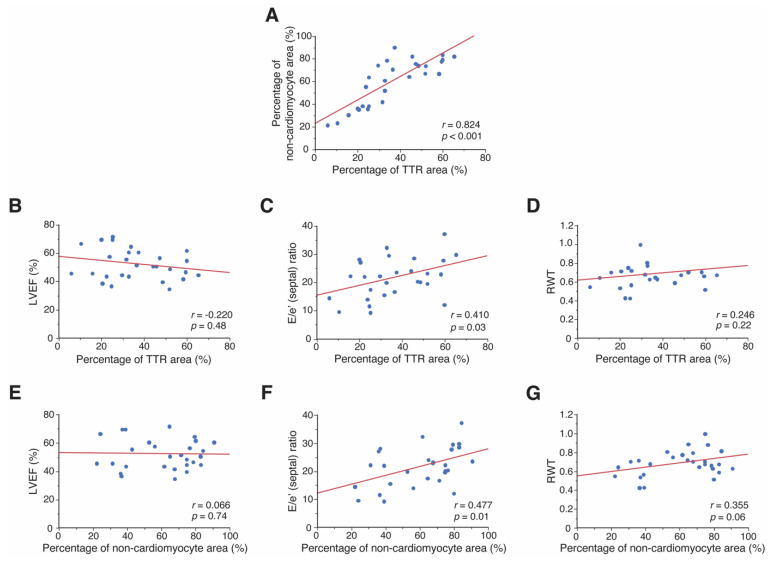
Pathological features and their association with echocardiographic parameters in patients with ATTR-CM. Relationship between the percentage of TTR area and (**A**) percentage of non-cardiomyocyte area, (**B**) LVEF, (**C**) E/e’ (septal) ratio, and (**D**) RWT. Relationship between the portion of non-cardiomyocyte area and (**E**) LVEF, (**F**) E/e’ (septal) ratio, and (**G**) RWT. ATTR-CM, transthyretin amyloid cardiomyopathy; E/e’ (septal) ratio, early transmitral filling velocity (E)/peak early diastolic velocity of the septal mitral annulus (e’) ratio; LVEF, left ventricular ejection fraction; RWT, relative wall thickness; TTR, transthyretin.

**Figure 3 jpm-12-00792-f003:**
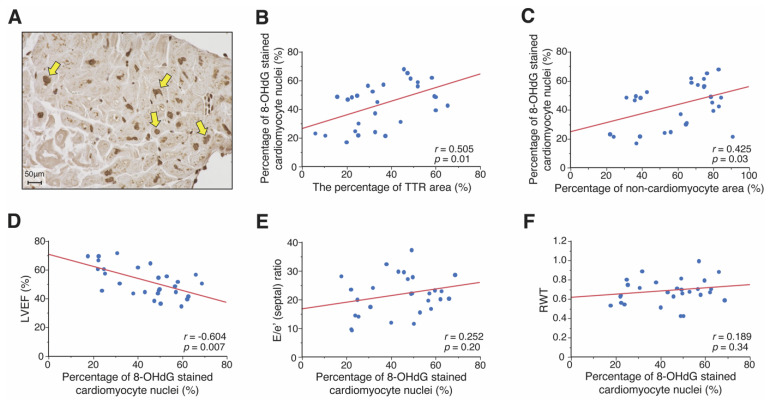
The association between immunostaining of 8-hydroxy-2′-deoxyguanosine (8-OHdG) and echocardiographic parameters in patients with ATTR-CM. (**A**) Immunostaining for 8-OHdG, visualized by diaminobenzidine (brown). 8-OHdG was found to be expressed in cardiomyocyte nuclei (Arrowheads). The relationship between the percentage of 8-OHdG stained cardiomyocyte nuclei and (**B**) percentage of TTR area, (**C**) the percentage of non-cardiomyocyte area, (**D**) LVEF, (**E**) E/e’ (septal) ratio, and (**F**) RWT. ATTR-CM, transthyretin amyloid cardiomyopathy; E/e’ (septal) ratio, early transmitral filling velocity (E)/peak early diastolic velocity of the septal mitral annulus (e’) ratio; LVEF, left ventricular ejection fraction; RWT, relative wall thickness; TTR, transthyretin.

**Figure 4 jpm-12-00792-f004:**
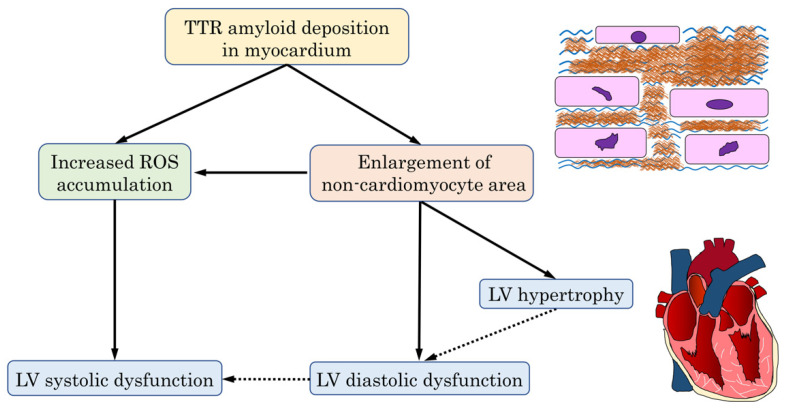
Schematic representation of the mechanism of cardiac dysfunction in ATTR-CM. In hearts displaying ATTR-CM, increased TTR amyloid deposition might be involved in the enlargement of the non-cardiomyocyte area, resulting in LV hypertrophy and LV diastolic dysfunction. Furthermore, the increased TTR amyloid deposition and subsequent enlargement of the non-cardiomyocyte area might lead to increased ROS accumulation, thus causing LV systolic dysfunction. ATTR-CM, transthyretin amyloid cardiomyopathy; LV, left ventricular; ROS, reactive oxygen species; TTR, transthyretin.

**Table 1 jpm-12-00792-t001:** Baseline clinical characteristics of the patients with ATTR-CM.

Characteristic	ATTR-CM (*n* = 28)
Age, years	76.2 ± 6.4
Male sex, *n* (%)	26 (92.9%)
**Comorbidities**	
Hypertension, *n* (%)	16 (57.1%)
Diabetes Mellitus, *n* (%)	10 (35.7%)
**Oral Medications**	
A CE-I/ARB, *n* (%)	18 (64.3%)
β-blocker, *n* (%)	13 (46.4%)
Loop diuretic, *n* (%)	17 (60.7%)
Mineralocorticoid receptor antagonists, *n* (%)	10 (35.7%)
**Electrocardiography**	
Low QRS voltage in limb leads, *n* (%)	16 (57.1%)
Poor R-wave progression in precordial leads, *n* (%)	15 (53.6%)
Atrial fibrillation, *n* (%)	13 (46.4%)
**Patterns of LV geometry**	
concentric LV hypertrophy, *n* (%)	27 (96.4%)
concentric LV remodeling, *n* (%)	1 (3.6%)
**Transthoracic echocardiography**	
LV ejection fraction, %	52.4 ± 10.6
LV end-diastolic diameter, mm	44.5 ± 5.6
LV end-systolic diameter, mm	34.7 ± 9.4
Intraventricular septal thickness, mm	15.6 ± 2.7
Posterior wall thickness, mm	15.1 ± 2.3
Relative wall thickness	0.69 ± 0.13
LV mass index, g/m^2^	175.6 ± 45.5
E/e’ (septal) ratio	21.7 ± 7.1
**Laboratory parameters**	
Hemoglobin, g/dL	12.7 ± 2.1
Uric acid, mg/dL	6.4 ± 2.1
eGFR, mL/min/1.73 m^2^	47.3 ± 15.7
Sodium, mEq/L	139.4 ± 2.3
Potassium, mEq/L	4.4 ± 0.5
Brain natriuretic peptide, pg/mL	298.1 (202.2, 533.9)
High-sensitivity cardiac troponin T, ng/L	72.5 (41.5, 95.0)
Plasma renin activity, µg/mL/h *	5.4 (4.1, 10.0)
Plasma aldosterone concentration, pg/mL ^†^	200.7 ± 183.6

Values are mean ± standard deviation, median (25th, 75th percentile), or number of patients (%). * The normal range of plasma renin activity is 0.3–3.9 ng/mL/h. ^†^ The normal range of plasma aldosterone concentration is 29.9–159.0 pg/mL. ACE-Is/ARBs, angiotensin-converting enzyme inhibitors/angiotensin II receptor blockers; ATTR-CM, transthyretin amyloid cardiomyopathy; E/e’ (septal) ratio, early transmitral filling velocity (E)/peak early diastolic velocity of the septal mitral annulus (e’) ratio; eGFR, estimated glomerular filtration rate; LV, left ventricular.

**Table 2 jpm-12-00792-t002:** Measurements of pathological features of the patients with ATTR-CM.

Characteristic	ATTR-CM (*n* = 28)
Percentage of the TTR area	36.2 ± 16.4 (%)
Percentage of the non-cardiomyocyte area	60.2 ± 20.5 (%)
Percentage of the cardiomyocyte area	39.8 ± 20.5 (%)

Values are mean ± standard deviation. ATTR-CM, transthyretin amyloid cardiomyopathy; TTR, transthyretin.

## Data Availability

All data supporting the findings are presented in the manuscript.

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
