# Peer review of "Transthyretin Amyloid Cardiomyopathy: Impact of Transthyretin Amyloid Deposition in Myocardium on Cardiac Morphology and Function"

_jpm, 2022, doi:10.3390/jpm12050792_

Round 1

Reviewer 1 Report

Nakano et al. conducted an interesting study on the endomyocardial biopsies of 28 patients with TTR amyloid cardiomyopathy (ATTR-CM) to investigate the association between echocardiographic parameters of left ventricular geometry and function and the extent of TTR plaques and (possibly) fibrosis on histological samples. They also assessed the potential role of oxidative stress on left ventricular geometry and function. Overall, the paper is well written, but I do believe that some points need further clarification:

  • The Authors apparently performed Masson’s trichrome staining. Why was the real extent of fibrosis not calculated on these slices?
  • Please provide a Table reporting the mean/median value + SD/IQR of the percentage of: TTR area, non-cardiomyocyte area, non-cardiomyocyte area without TTR, and cardiomyocyte area (total – non-cardiomyocyte).
  • Please provide correlations between the percentage of cardiomyocyte area and LVEF, E/e’, RWT. Was there any significant association with RWT?
  • Given that most patients had preserved LVEF, a correlation analysis between histological parameters and echocardiographic indices of subclinical systolic dysfunction would significantly improve the paper (e.g., global longitudinal strain). If not available, I would emphasize this point as a limitation of the study.
  • How many EMB samples were collected from each patient? How many EMB samples were analysed with histology for each patient?
  • Were patients taking any ATTR-CA disease-modifying medications (e.g., tafamidis and so on)?
  • The Authors should underscore as a limitation of the paper the fact that genetic analysis was performed only in a subgroup of patients.
  • It’s a bit odd that the percentage of non-cardiomyocyte area tended to correlate with RWT while its individual components (TTR are and non-TTR area) did not show any association with RWT. Please comment on this.
  • A structured abstract (background, aims, methods, results, conclusion) would be preferable.
  • Abstract and introduction: “Transthyretin (TTR) amyloid cardiomyopathy (ATTR-CM) has recently been recognized as a cause of left ventricular (LV) hypertrophy (LVH) and progressive heart failure in elderly patients”. ATTR-CM has been known for decades as a cause of LVH and HF. Perhaps the Authors meant that ATTR-CM is increasingly recognized as a cause of LVH and HF, right?
  • Please rewrite this sentence (introduction) to clarify what you mean: “Although ATTR-CM has been reported to manifest as concentric LVH [4], it is not clear whether this geometry is applicable to ATTR-CM cases in general.”
  • Table 1, please report hsTnT in ng/L.
  • Figure 1. When citing the ASE guidelines, please report the corresponding reference.
  • Discussion: “This study suggests that the transition of geometry in patients 268 with ATTRwt-CM was consistent with that in ATTRv-CM patients”. The problem with this sentence is that the Authors assume that all the patients included in the present study had ATTRwt, which cannot be proved unless the genetic analysis is performed on the whole cohort. Please rephrase this concept and be clearer about the limitations of the paper.
  • When discussing BNP values and ATTR-CM disease stage, I suggest citing the following paper: doi.org/10.1002/ejhf.2113.
  • When writing about the role of oxidative stress in infiltrative cardiomyopathies and amyloidosis, I suggest citing the following papers: doi.org/10.1161/CIRCRESAHA.121.318187; 10.1177/2047487319870344.

Author Response

We thank the reviewer for their thoughtful comments on our manuscript. Along the lines suggested by the reviewer, we have revised the manuscript. Added or corrected sentences according to the reviewers’ comments have made all changes in red and bold font in the main text.

Point 1: The Authors apparently performed Masson’s trichrome staining. Why was the real extent of fibrosis not calculated on these slices?

 Response 1: Thank you for the reviewer’s important suggestions. Masson’s trichrome stained slice and TTR immunostained slice were not continuous slices. Therefore, the interstitial fibrosis region measured using Masson's trichrome staining was considered to be inconsistent with the interstitial area measured using the TTR immunostained slice. Furthermore, by measuring pathological features including the interstitial region and the TTR area in the same TTR immunostaining slice, we were able to accurately and rationally evaluate the relationship between those parameters. Therefore, in this study, we had measured the interstitial area using the TTR immunostained slice.

Point 2: Please provide a Table reporting the mean/median value + SD/IQR of the percentage of: TTR area, non-cardiomyocyte area, non-cardiomyocyte area without TTR, and cardiomyocyte area (total – non-cardiomyocyte).

Response 2: Thank you for the reviewer’s appropriate suggestions. As reviewer pointed out, we added the measurements of phathological features in Results and Table 2 as below. Here, we have omitted the non-cardiomyocyte area without TTR as described later.

  Results (Page 5, line 192-193); The measurements of the degree of the area that stained for TTR, the non-cardiomyocyte area and the cardiomyocyte area were represented in Table 2.

Results (Page 5, line 194); Table 2. Measurements of pathological features of the patients with ATTR-CM.

Characteristic

ATTR-CM (n = 28)

Percentage of the TTR area

36.2 ± 16.4 (%)

Percentage of the non- cardiomyocyte area

60.2 ± 20.5 (%)

Percentage of the cardiomyocyte area

39.8 ± 20.5 (%)

Values are mean ± standard deviation. TTR, transthyretin.

Point 3: Please provide correlations between the percentage of cardiomyocyte area and LVEF, E/e’, RWT. Was there any significant association with RWT?

Response 3: Thank you for the reviewer’s appropriate suggestions. We assessed the correlation between the percentage of cardiomyocyte area and LVEF, E/e’and RWT. As shown in the figure below, the percentage of cardiomyocyte area was negatively correlated with E/e' (p = 0.01), and tended to have a negative correlation with RWT (p = 0.06). Since the percentage of non-cardiomyocyte area is the value obtained by subtracting one from the percentage of cardiomyocyte area, the analysis result of the correlation between the percentage of cardiomyocyte area or the percentage of non-cardiomyocyte area and echocardiographic parameters were substantially same. Therefore, we have presented only the correlation between the percentage of non-cardiomyocyte area and echocardiographic parameters in this manuscript.

Point 4: Given that most patients had preserved LVEF, a correlation analysis between histological parameters and echocardiographic indices of subclinical systolic dysfunction would significantly improve the paper (e.g., global longitudinal strain). If not available, I would emphasize this point as a limitation of the study.

Response 4: Thank you for the reviewer’s essential suggestions. As the reviewer pointed out, most patients enrolled in this study had preserved LVEF. Unfortunately, echocardiographic parameters of subclinical systolic dysfunction, such as global longitudinal strain, were measured in only four patients. The samples size was too small to perform a stastical calculation. As reviewer pointed out, we added the sentences in Limitation as follow;

  Discussion (Page9, line 327-335); Fourthly, echocardiographic parameters of subclinical systolic dysfunction, such as global longitudinal strain, were measured in only four patients in this study. Given that the most patients with ATTR-CM had preserved LVEF, further investigation using speckle tracking analysis is needed to elucidate the relation between the TTR deposition and cardiac dysfunction. Fifthly, the endomyocardial biopsy technique may have sampling errors in establishing a pathological diagnosis, although endomyocardial biopsy is reported to have a higher positive probability of detecting amyloid deposition than other methods, such as abdominal fat tissue biopsy in ATTR-CM patients.

Point 5: How many EMB samples were collected from each patient? How many EMB samples were analysed with histology for each patient?

Response 5: Thank you for the reviewer’s important suggestions. We usually collect one to two endomyocardial biopsy samples from each patient. In this study, we analyzed pathological features using one to two endomyocardial biopsy samples for each patient.

Point 6: Were patients taking any ATTR-CA disease-modifying medications (e.g., tafamidis and so on)?

Response 6: Thank you for the reviewer’s appropriate suggestions. All patients enrolled in this study did not take any ATTR-CM disease-modifying medications. As reviewer pointed out, we added a sentence in Results as follow;

 Results (Page 4, line 168-169); All patients had taken neither stabilizers of TTR homotetramers nor RNA interference therapies inhibiting TTR synthesis. No mutations in the TTR gene were found in 17 consenting patients.

Point 7: The Authors should underscore as a limitation of the paper the fact that genetic analysis was performed only in a subgroup of patients.

Response 7: Thank you for the reviewer’s essential suggestion. Because of retrospective study, we performed genenitic analysis of the TTR gene in only 17 patients. As reviewer pointed out, we added a sentence in Limitation as follow;

  Discussion (Page 9, line 327); Third, not all patients with ATTR-CM underwent genetic analysis.

Point 8: It’s a bit odd that the percentage of non-cardiomyocyte area tended to correlate with RWT while its individual components (TTR are and non-TTR area) did not show any association with RWT. Please comment on this.

Response 8: We apologize to the reviewer for confusion. Since we evaluated using TTR immunostained slices, we measured the TTR-positive interstitial area as the TTR area and the TTR-negative interstitial area as the non-cardiomyocyte area without TTR. However, when observing the endomyocardial biopsy samples using an electron microscope, the amyloid fibrils coexist with the collagen fibrils in the extracellular region. Therefore, collagen fibers are present not only in the non-cardiomyocyte area without TTR (the non-cardiomyocyte area minus the TTR area) but also in the TTR area. As a result of re-evaluation from the reviewer's comment, we considered that it was appropriate to express the extracellular region by the non-cardiomyocyte area that combines the TTR area and the non-cardiomyocyte without TTR, rather than dividing the interstitial area into those two areas. As the original manuscript can be confusing to the readers as well as to the reviewer, we have removed the analysis using the non-cardiomyocyte area without TTR, and then we analyzed using the non-cardiomyocyte area that more accurately reflected the entire interstitial region. From those findings, we revised and added the sentences and Figures as follows;

  Methods (Page 4, line 149-152); The non-cardiomyocyte area, indicated the extracellular region, was calculated as the difference between the total myocardial area and cardiomyocyte area. In addition, the non-cardiomyocyte area that did not contain TTR indicated the area of interstitial fibrosis; so this fibrotic area was determined from the difference between the non-cardiomyocyte and TTR areas. The non-cardiomyocyte area and the area containing TTR, and the non-cardiomyocyte area without TTR, were calculated as percentages of the total myocardial area.

Results (Page 6, line 197-205); However, the proportion that was TTR positive did not correlate with the non-cardiomyocyte area that did not exhibit TTR (which is almost the area of interstitial fibrosis), nor the cardiomyocyte diameter (Figure 2B and Supplementary Figure S3A). The size of the TTR area correlated with the E/e’ (septal) ratio (p = 0.03; Figure 2C), but not with LVEF or RWT, which is a marker of LVH (Figure 2B, and 2D). Although the extent of the non-cardiomyocyte area was not associated with LVEF (Figure 2E), it was positively correlated with the E/e’ (septal) ratio (p = 0.01; Figure 2F) and tended to a positive correlation with RWT (p = 0.06; Figure 2G). Furthermore, the extent of the non-cardiomyocyte area without TTR was not correlated with LVEF, E/e’ (septal) ratio, or RWT (Figure 2I, 2J, and 2K). Cardiomyocyte diameter was not correlated with the size of the non-cardiomyocyte area, or the non-cardiomyocyte area without TTR (Supplementary Figure S3B and S3C), or the LVEF, E/e’ (septal) ratio, or the RWT (Supplementary Figure S3B, S3D, S3E, and S3F).

Results (Page 7, line 219-221); The fraction of 8-OHdG stained cardiomyocyte nuclei was positively correlated with the percentage of the non-cardiomyocyte area and cardiomyocyte diameter (p = 0.03, and p = 0.04, respectively; Figure 3C and Supplementary Figure S3C), although it was not cor-related with that of the non-cardiomyocyte area not displaying TTR (Figure 3D).

Discussion (Page 8, line 274-275); Our study suggested that TTR amyloid deposition was significantly associated with the extent of the non-cardiomyocyte area. However, it was not correlated with the degree of interstitial fibrosis. These results suggest that the non-cardiomyocyte area was enlarged mainly by increased TTR amyloid deposition.

Figure 2;

Figure 2 Legend (Page 6 line 207-210); Pathological features and their association with echocardiographic parameters in ATTR-CM patients. Relationship between the percentage of TTR area and (A) percentage of non-cardiomyocyte area, (B) percentage of non-cardiomyocyte area without TTR, (B) LVEF, (C) E/e’ (septal) ratio, and (D) RWT. Relationship between the portion of non-cardiomyocyte area and (E) LVEF, (F) E/e’ (septal) ratio, and (G) RWT. Relationship between the portion of non-cardiomyocyte area without TTR and (I) LVEF, (J) E/e’ (septal) ratio, and (K) RWT.

Figure 3;

Figure 3 Legend (Page 7, line 228-230); The relationship between the percentage of 8-OHdG stained cardiomyocyte nuclei and (B) per-centage of TTR area, (C) the percentage of non-cardiomyocyte area, (D) percentage of non-cardiomyocyte area without TTR deposition, (D) LVEF, (E) E/e’ (septal) ratio, and (F) RWT.

Point 9: A structured abstract (background, aims, methods, results, conclusion) would be preferable.

Response 9: Thank you for the reviewer’s crusial suggestion. As reviewer pointed out, we revised to a structured abstract as follow;

  Abstract (Page 1, line 21-23); Background: Recently, transthyretin (TTR) amyloid cardiomyopathy.       (ATTR-CM) is increasingly recognized as a cause of left ventricular (LV) hypertrophy (LVH) and progressive heart failure in elderly patients.

  Abstract (Page 1, line 25-28); Methods: We studied 28 consecutive patients with ATTR-CM and analyzed the relationship between echocardiographic parameters and pathological features using endomyocardial biopsy samples. Results: The cardiac geometries of patients with ATTR-CM were mainly classified as concentric LVH (96.4%).

  Abstract (Page 1, line 34-36); Conclusion: Patients with ATTR-CM mainly displayed concentric LVH. geometry.

Point 10: Abstract and introduction: “Transthyretin (TTR) amyloid cardiomyopathy (ATTR-CM) has recently been recognized as a cause of left ventricular (LV) hypertrophy (LVH) and progressive heart failure in elderly patients”. ATTR-CM has been known for decades as a cause of LVH and HF. Perhaps the Authors meant that ATTR-CM is increasingly recognized as a cause of LVH and HF, right?

Response 10: Thank you for the reviewer’s important suggestion. What we meant was as reviewer pointed out. We revised the sentences in Abstract and Introduction as follows;

  Abstract (Page 1, line 21-23); Recently, transthyretin (TTR) amyloid cardiomyopathy (ATTR-CM) is increasingly recognized as a cause of left ventricular (LV) hypertrophy (LVH) and progressive heart failure in elderly patients.

  Introduction (Page 2, line 52-54); Recently, ATTRwt-CM is increasingly recognized as a cause of. progressive heart failure, restrictive cardiomyopathy, or LVH in elderly patients [2, 5, 6, 8].

Point 11: Please rewrite this sentence (introduction) to clarify what you mean: “Although ATTR-CM has been reported to manifest as concentric LVH [4], it is not clear whether this geometry is applicable to ATTR-CM cases in general.”

Response 11: We apologize to the reviewer for confusion. As reviewer pointed out, we revised the sentence in the Introduction as follow;

  Introduction (Page 2, line 59-61); Although ATTR-CM has been reported to manifest as concentric LVH [4], it is not clear whether this geometry is applicable to a greater number of cases with biopsy proven ATTR-CM.

Point 12: Table 1, please report hsTnT in ng/L.

Response 12: Thank you for the reviewer’s important suggestion. As reviewer pointed out, we revised the unit of hsTnT from ng/mL to ng/L. We revised a sentence in Results and Table 1 as follow;

  Results (Page 4, line 167-168); The median brain natriuretic peptide (BNP) level was 298.1 pg/mL and the median high-sensitivity cardiac troponin T was 72.5 ng/L.

Table 1;

Characteristic

ATTR-CM (n = 28)

High-sensitivity cardiac troponin T, ng/L

72.5 (41.5, 95.0)

Point 13: Figure 1. When citing the ASE guidelines, please report the corresponding reference.

Response 13: Thank you for the reviewer’s important suggestion. As reviewer pointed out, we added the corresponding reference in the Figure 1 legend.

  Results (Page 5, line 183-185); Figure 1. The figure shows the distribution of patient results according to. the four geometries of the American Society of Echocardiography [23]: (A) concentric LVH, (B) eccentric LVH, (C) con-centric LV remodeling, and (D) normal.

Point 14: “This study suggests that the transition of geometry in patients with ATTRwt-CM was consistent with that in ATTRv-CM patients”. The problem with this sentence is that the Authors assume that all the patients included in the present study had ATTRwt, which cannot be proved unless the genetic analysis is performed on the whole cohort. Please rephrase this concept and be clearer about the limitations of the paper.

Response 14: Thank you for the reviewer’s important suggestion. As reviewer pointed out, not all patients underwent genetic analysis, so it was not possible to evaluate the transition of geometry in patients with ATTRwt-CM in the present study. We revised and added a sentence in Discussion as follow;

Discussions (Page 8, line 262-263); The present study suggests that the transition of geometry in patients with ATTR-CM was consistent with that in ATTRv-CM patients.

Point 15: When discussing BNP values and ATTR-CM disease stage, I suggest citing the following paper: doi.org/10.1002/ejhf.2113.

Response 15: Thank you for the reviewer’s important suggestion. As reviewer pointed out, we cited the paper the reviewer kindly presented.

Point 16: When writing about the role of oxidative stress in infiltrative cardiomyopathies and amyloidosis, I suggest citing the following papers: doi.org/10.1161/CIRCRESAHA.121.318187; 10.1177/2047487319870344.

Response 16: Thank you for the reviewer’s important suggestion. As reviewer pointed out, we cited the papers the reviewer kindly presented.

Please see attachment below.

Reviewer 2 Report

May I congratulate the authors on this well constructed manuscript, detailing a robust study of how echocardiographic parameters of  LV systolic and diastolic function are corelated with the amyloid burden, using EMB - anatomo-pathology., in ATTR cardiomyopathy. The authors clearly highlights that the presence of f TTR deposition was positively correlated with enlargement of  the non-cardiomyocyte area, and these were positively correlated with LV diastolic dysfunction. Moreover, ROS accumulation in cardiomyocytes was positively correlated with LV systolic dysfunction. The manuscript is well written, the concepts are clearly exposed and the figures are very clear and representative. The methodology is generally very well detailed and results are clearly stated. 

Author Response

Thank you so much for understanding the efforts we made for this study and for evaluating appropriately.